# Giga-scale Kernel Matrix-Vector Multiplication on GPU

**Robert Hu**
Amazon *
robyhu@amazon.co.uk

**Siu Lun Chau**
Department of Statistics
University of Oxford
siu.chau@stats.ox.ac.uk

**Dino Sejdinovic**
School of Computer and Mathematical Sciences
University of Adelaide *
dino.sejdinovic@adelaide.edu.au

**Joan Alexis Glaunès**
MAP5
Université Paris Descartes
alexis.glaunes@mi.parisdescartes.fr

## Abstract

Kernel matrix-vector multiplication (KMVM) is a foundational operation in machine learning and scientific computing. However, as KMVM tends to scale quadratically in both memory and time, applications are often limited by these computational constraints. In this paper, we propose a novel approximation procedure coined *Faster-Fast and Free Memory Method* ($F^3M$) to address these scaling issues of KMVM for tall ($10^8 \sim 10^9$) and skinny ($D \leq 7$) data. Extensive experiments demonstrate that $F^3M$ has empirical *linear time and memory* complexity with a relative error of order $10^{-3}$ and can compute a full KMVM for a billion points *in under a minute* on a high-end GPU, leading to a significant speed-up in comparison to existing CPU methods. We demonstrate the utility of our procedure by applying it as a drop-in for the state-of-the-art GPU-based linear solver FALKON, *improving speed 1.5-5.5 times* at the cost of $< 1\%$ drop in accuracy. We further demonstrate competitive results on *Gaussian Process regression* coupled with significant speedups on a variety of real-world datasets.

## 1 Introduction

Kernel matrix-vector multiplication (KMVM) is one of the most important operations needed in scientific computing with core applications in diffeomorphic registration, geometric learning [11], [31], numerical analysis [28], fluid dynamics [6], and machine learning [27]. For a dataset of size $n$, KMVM using direct computation has complexity and memory footprint $\mathcal{O}(n^2)$, both unfeasible for modern large scale applications where $n \approx 10^9$ is becoming increasingly common. Pioneering contributions presented in the *Fast Multipole Method* (FMM) [10] amend the complexity of these problems to $\mathcal{O}(n \log{(\epsilon^{-1})})$, where $\epsilon$ is the chosen error tolerance, with varying reductions in memory footprint for data restricted to dimension $D = 2$. Subsequent developments in [8, 17] mainly focused on extending approximations for a broader set of kernels for a fixed dimensionality $D \leq 3$, tailored for problems in physics with narrow data such as electrostatics, stellar dynamics, Stokes flow, and acoustic problems, amongst others.

---

*Work mainly done while the authors were with the Department of Statistics, University of Oxford.

36th Conference on Neural Information Processing Systems (NeurIPS 2022).

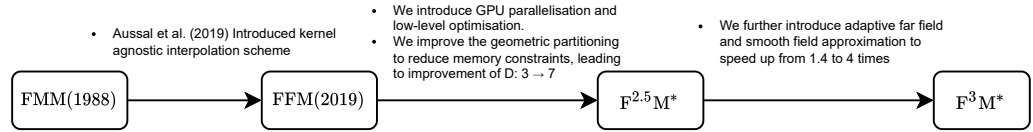

Figure 1: A brief summary of the evolution of the FMM family. Our contributions in *.

In this paper, we introduce *Faster-Fast and Free Memory Method* ($F^3M$), a novel algorithm built upon the FFM [3] framework to perform KMVM on a GPU for *tall and skinny* ($D \leq 7$) data of order $n \sim 10^9$ in under a minute with user-specified error tolerance, providing between $2 - 8500$ times speed-up over existing methods. It should be noted that the constraints on $D$ and $n$ are not inherent formal constraints, but a reflection of practical limits with typical current computational resources.

**Notations.** We use capital and lower case bold letters to represent matrices and vectors, respectively. In this paper, we will work with matrices $\mathbf{X} \in \mathbb{R}^{n_x \times D}$, $\mathbf{Y} \in \mathbb{R}^{n_y \times D}$ and vector $\mathbf{b} \in \mathbb{R}^{n_y}$. For a kernel $k$, the goal for KMVM is to compute $\mathbf{v} := \mathbf{K} \cdot \mathbf{b}$, where $\mathbf{K} := k(\mathbf{X}, \mathbf{Y}) = \{k(\mathbf{x}_i, \mathbf{y}_j)\}_{i=1, j=1}^{n_x, n_y}$, and $\mathbf{x}_i, \mathbf{y}_j$ denote the $i^{th}, j^{th}$ row of $\mathbf{X}, \mathbf{Y}$ respectively.

## 2   Motivation and Related Work

Kernel methods are often limited by their $\mathcal{O}(n^2)$ memory footprint and computational complexity for KMVM. These constraints make scaling beyond $n = 10^6$ challenging. Many recent developments have been made to improve both of these constraints, ranging from hardware acceleration using GPUs in KeOps [11], to various approximation techniques proposed in [36, 33, 34, 3, 9]. In this work, we focus our attention on kernel independent KMVM methods.

**KeOps.** Charlier et al. [11] proposes a map-reduce scheme to compute kernels using exactly $\mathcal{O}(n)$ memory and $\mathcal{O}(n^2)$ complexity on GPU. This is achieved by computing the full KMVM product on-the-fly by summing $v_i = \sum_{j=1}^{n} k(\mathbf{x}_i, \mathbf{y}_j) b_j$ directly, without ever storing the kernel matrix $\mathbf{K} \in \mathbb{R}^{n \times n}$ explicitly. Extensive experiments show that this method is practical when $n \leq 10^6$, as the GPU hardware acceleration allows the KMVM product to be computed in less than a second on a conventional GPU. Moreover, the method places no constraint on the number of features $D$ it can be applied to, making it favourable for KMVM on medium size datasets. In application contexts, KeOps is currently adopted into conjugate gradient solver FALKON [24, 26] as part of the default pipeline.

[2]

**The Fast and Free Memory Method (FFM).** While KeOps can theoretically scale to a billion points, it becomes practically infeasible as the $\mathcal{O}(n^2)$ complexity would imply a computational time of $10^6$ seconds, or roughly 11 days. To overcome this billion points barrier, Aussal et al. [3] deploys a geometric space partitioning scheme, and proposed the *Fast and Free Memory Method* (FFM), a KMVM approach that extends the FMM [10] family of algorithms. In contrast to traditional FMM methods, which require specific series expansion of the kernel, FFM deploys Lagrange interpolations to approximate them instead. This allows FFM to be applied to almost any conventional kernel and further enables the user to trade off accuracy with computational efficiencies by controlling the order of the approximating polynomial [19]. Compared to KeOps, FFM demonstrates both linear memory and time complexity in experiments and scales to compute a billion-points KMVM on a smaller CPU cluster under 4 hours, outscaling the GPU implementation of FMM [21]. While 4 hours is a significant improvement compared to 11 days from KeOPS, it still renders many machine learning techniques infeasible. Further, as recursive partitioning of the data space scales poorly with $D$ [5], both FMM and FFM can only be applied to $D \leq 3$ data, a price to pay for the speed-up of KMVM operations when $n = 10^9$. Furthermore, we show in our experiments that a direct FFM port to GPU gives unstable results for $n = 10^9$, $D = 3$ for non-trivial data simulations (bottom row in Appendix 10).

Table 1: Comparison between methods(* indicate ours).

| Method | FMM | KeOps | FFM | $F^{2.5}M^*$ | $F^3M^*$ |
|---|---|---|---|---|---|
| Kernel Independent | | ✓ | ✓ | ✓ | ✓ |
| Linear Time | ✓ | | ✓ | ✓ | ✓ |
| Linear Memory | | ✓ | ✓ | ✓ | ✓ |
| Restriction in $D$ | $\leq 3$ | | $\leq 3$ | $\leq 7$ | $\leq 7$ |
| GPU | ✓ | ✓ | | ✓ | ✓ |
| Scales to $n = 10^9$ under 1 hour [1] | | | | ✓ | ✓ |
| under 1 minute! [1] | | | | | ✓ |

---

[2]KMVM applied to 3D data for $n = 10^9$ on a Nvidia V100 GPU.

**Our contribution.** To surpass the billion point barrier while maintaining high-speed and stable computation, we propose $F^{2.5}M$ and our main algorithm $F^3M$, the first pair of KMVM algorithms that can reliably scale to $n = 10^9$ on skinny data using a single GPU. We build $F^{2.5}M$ on top of FFM by introducing non-trivial GPU parallelisation and low-level optimisations. We further stabilize and improve the original geometric partitioning scheme in FFM to significantly reduce memory constraints, leading to a relaxation of dimensionality constraints from 3 to 7. At last, we introduce an adaptive far-field and smooth field approximation scheme for kernel interpolation, resulting in our main algorithm $F^3M$, which runs $2.0 - 33.3$ times quicker and more stable than a direct port of FFM on GPU. See Fig. 1 and Table 1 for an overview and comparisons of the methods. We summarise our contribution as follows:

**1.** We propose *Faster-Fast and Free Memory Method* ($F^3M$), a KMVM algorithm building on top of FFM by applying multiple low-level enhancements, GPU parallelisation, and algorithmic computational and memory enhancements, allowing for KMVM operations on $n \leq 10^9$ data in under a minute. Codebase is released here [14].

**2.** We characterize theoretical time and memory complexity of $F^3M$.

**3.** We run extensive KMVM experiments of $F^3M$ on a variety of tall and skinny data with $n \leq 10^9$, demonstrating empirical linear time and memory scaling, and achieving speedups between 2–8500 times when compared to FFM (GPU and CPU) and KeOps.

**4.** We run a practical application of $F^3M$ as a drop-in replacement for KeOps in conjugate gradient solver FALKON [24, 26] for kernel ridge regression and classification (KRR) on giga-scale data, obtaining a solution 3.4 times faster with <1% drop in accuracy. We further demonstrate competitive results on Gaussian process regression against KISS-GP [35], SVGP [18] and SVGR [30] with significant speed-ups.

## 3 Background

The FFM method considers KMVM for a kernel $k$ evaluated on two data matrices $\mathbf{X}, \mathbf{Y}$ and $\mathbf{b}$ are weights associated with $\mathbf{Y}$. The KMVM is expressed as $\mathbf{v} := k(\mathbf{X}, \mathbf{Y}) \cdot \mathbf{b} = \mathbf{K} \cdot \mathbf{b}$. For example, $\mathbf{b}$ could be the weights in a KRR or the strength of electronic charges. As $n_x$ and $n_y$ are taken to be very large, a full computation is unfeasible. In this section, we illustrate and detail the main steps of FFM, before presenting our improvements in Section 4.

For illustration purposes, we first consider a simple 2D KMVM. Our goal is to calculate $k(\mathbf{X}, \mathbf{Y}) \cdot \mathbf{b}$ for $\mathbf{X}, \mathbf{Y}$ in Figure 2. The intuition behind FFM is to reduce the complexity of calculating the full KMVM by partitioning $\mathbf{X}$ and $\mathbf{Y}$ such that certain calculations can be approximated in a fast manner, based on the pairwise distances between partitions.

**Enclosing and partitioning the data.** The first step is to partition the data. To begin, we find a large enough box that can just enclose $\mathbf{X}$ or $\mathbf{Y}$. The edge length of this box is calculated as

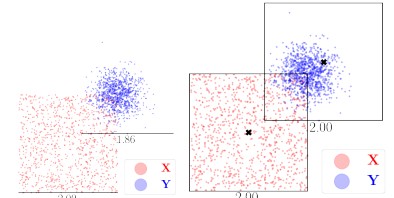

Figure 2: Enclosing $\mathbf{X}$ and $\mathbf{Y}$ within a box, "x" marks the center of the box. Numbers under the the boxes denotes edge length. In the right plot, we have enclosed the blue points with the largest box.

$$\mathcal{E} := \max\left(\max_d(x_{\max}^{(d)} - x_{\min}^{(d)}), \max_d(y_{\max}^{(d)} - y_{\min}^{(d)})\right)$$

where $x_{\max}^{(d)}, x_{\min}^{(d)}$ denotes the largest value and the smallest value along the $d$-dimension in $\mathbf{X}$ and similarly for $\mathbf{Y}$. Figure 2 illustrates this enclosing procedure.

**Defining near and far-field** In FMM, an octree [23] is applied to recursively partition data into

smaller boxes $B_p^X \subset \mathbf{X}, B_q^Y \subset \mathbf{Y}$, with $p, q$ denoting box indices. Here each box corresponds to a subset of rows in the data matrix. Let us also denote $\mathbf{b}_q$ as the partition of $b_j$'s grouped with the same indices as $B_q^Y$. To calculate the KMVM between two boxes $B_p^X, B_q^Y$ with the grouped vector $\mathbf{b}_q$, for each $\mathbf{x}_i \in B_q^X$, we com-

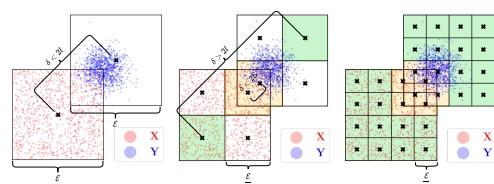

Figure 3: Recursive partitioning of $\mathbf{X}$ and $\mathbf{Y}$ for 2D data. Far-field interactions are colored green while near-field interactions are colored orange and $\delta$ denotes the euclidean distance between the centers.

pute

$$v_i^{p,q} = \sum_{\mathbf{y}_j \in B_q^Y, b_j \in \mathbf{b}_q} k(\mathbf{x}_i, \mathbf{y}_j) b_j \qquad (1)$$

with $\mathbf{v}^{p,q} = [v_1^{p,q} \dots v_{n_x}^{p,q}]$. Now the target $\mathbf{v}$ can be computed as $\mathbf{v} = \sigma([\mathbf{v}^{p=1}, \dots, \mathbf{v}^{p=P}]^\top)$, where $\mathbf{v}^p = \sum_{q=1}^Q \mathbf{v}^{p,q}$, $P, Q$ denote the total number of boxes and $\sigma(\cdot)$ a permutation such that $v_i^p$ appear in the same order as $\mathbf{x}_i$ appears in $\mathbf{X}$. Figure 3 shows how boxes are recursively partitioned.

**Far and near-field interactions** FFM relies on a divide-and-conquer strategy to effectively compute a KMVM product; data is partitioned into boxes and then separated into far-field and near-field interactions, where near-field interactions are computed exactly and far-field interactions are approximated using Lagrange interpolation for speed, explained in the paragraph below. The partitioning procedure in FFM is recursive, where the recursion depth `tree_depth` controls the size of the edge $l = \frac{\mathcal{E}}{2^{\text{tree\_depth}}}$ of the box. An interaction is defined to be in the far-field if the distance between the two center points of the boxes exceeds $2l$, i.e. `IsFarField` $:= \|\mathbf{x}_{\text{center}} - \mathbf{y}_{\text{center}}\| \geq 2l$. While $l$ for each box will decrease with the number of divisions, this rule ensures a fixed minimal distance for a given depth for far-field interactions. Figure 3 illustrates how far(green) and near(orange)-field interactions arise between $\mathbf{X}$ and $\mathbf{Y}$ when `tree_depth` increases.

**Lagrange interpolation** We review Lagrange interpolation used for far-field approximations in FFM. Given a function $f(x) : [-1, 1] \to \mathbb{R}$ and $r+1$ unique points $s_i \in [-1, 1]$, $i = 0, \dots, r$, there exists a unique polynomial $p_r(x)$ of degree $\leq r$ that interpolates $f$ at $p_r(s_i) = f(s_i)$. The Lagrange polynomial is given by $p_r(t) = \sum_{i=0}^r f(s_i)\mathcal{L}_i(t)$, where $\mathcal{L}_i(t) = \frac{\prod_{j=0, j\neq i}^r (t - s_j)}{\prod_{j=0, j\neq i}^r (s_i - s_j)}$, $i = 0, \dots, r$. We are free to chose the degree $r$ as well as the points $s_i$ to interpolate through. The choice of $s_i$ is especially important in minimizing large oscillations around the edges of the interpolation interval (Runge's phenomenon [12]). For this reason, Chebyshev nodes of the second kind are used [7] $s_i = \cos\theta_i$, where $\theta_i = \frac{i\pi}{r}$, $i = 0, \dots, r$.

**Interpolating** $k(\mathbf{x}, \mathbf{y})$ By noticing that $k(\mathbf{x}, \mathbf{y})$ is a bivariate function, we can apply Lagrange interpolation twice, thus interpolating $k(\mathbf{x}, \mathbf{y})$ as $k(\mathbf{x}, \mathbf{y}) \approx \sum_{i=1}^{r_X} \mathcal{L}_i(\mathbf{x}) \sum_{j=1}^{r_Y} k\left(\mathbf{s}_i^x, \mathbf{s}_j^y\right) \mathcal{L}_j(\mathbf{y})$. Here $r_X, r_Y$ denotes the number of the interpolation nodes and $\mathbf{s}_i^x, \mathbf{s}_j^y \in \mathbb{R}^D$ denotes the grid of interpolation nodes for $B_p^X$ and $B_q^Y$. Note that since $\mathbf{x} \in \mathbb{R}^D$, we take $\mathcal{L}_i(\mathbf{x}) :=$ $\prod_{d=1}^D \frac{\prod_{j=0, j\neq i}^r \left(x^{(d)} - s_j^{(d)}\right)}{\prod_{j=0, j\neq i}^r \left(s_i^{(d)} - s_j^{(d)}\right)}$, $i = 0, \dots, r$. These operations can be vectorized and computed se-

quentially on-the-fly with linear memory footprint $\mathbf{v} \approx \mathbf{L}_X^T \cdot \overbrace{(\mathbf{K} \cdot \underbrace{(\mathbf{L}_Y \cdot \mathbf{b})}_{\mathbf{v}_1})}^{\mathbf{v}_2}$, which is done by first

computing $\mathbf{v}_1$, then $\mathbf{v}_2$ and lastly $\mathbf{v}$. Here $\mathbf{L}_X$ denotes a matrix with entries $\mathcal{L}_i(\mathbf{x}_j)$, where $i$ indexes the rows and $j$ the columns, with $\mathbf{L}_Y$ following the same definition for $\mathbf{y}_j$'s instead. A far-field KMVM between two boxes $\mathbf{v}_{p,q} = k(B_p^X, B_q^Y) \cdot \mathbf{b}_q$ is then approximated by using double Lagrange interpolation according to Figure 4.

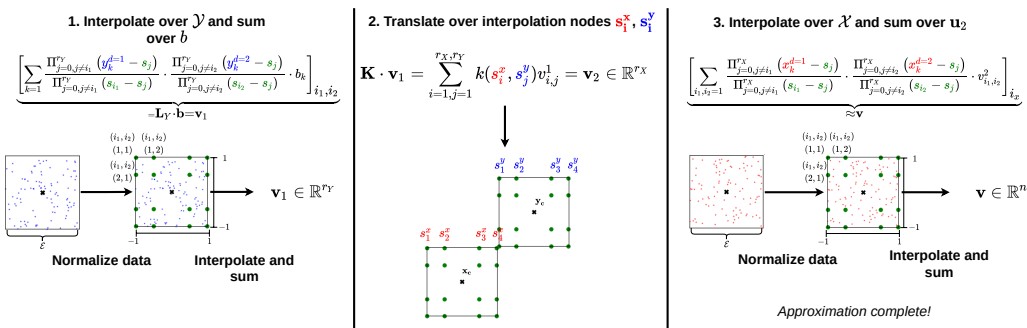

Figure 4: Here we approximate a far-field interaction between two boxes. In **1.** we first normalize the data between $[-1, 1]$ and conduct 2D interpolation of $k(\cdot, \mathbf{y})$ while summing over $\mathbf{b}$. In **2.**, the interpolation for $k(\mathbf{x}, \cdot)$ will also be normalized and hence we need to translate the distance between the boxes by calculating $\mathbf{K} \cdot \mathbf{v}_1$. Lastly in **3.** we interpolate $k(\mathbf{x}, \cdot)$ while we sum $\mathbf{v}_2$.

# 4 Faster-FFM ($\text{F}^{2.5}\text{M}$ and $\text{F}^3\text{M}$)

To fully leverage the port of FFM to GPU, we enhance FFM with novel approximation procedures for improved complexity and memory optimizations to scale to $n = 10^9$. We coin this improved version Faster-FFM ($\text{F}^3\text{M}$). The capabilities of $\text{F}^3\text{M}$ against previous methods are summarized in Table 1.

## 4.1 CPU to GPU optimizations

In FFM, every computation is serial and on CPU. When moving to GPU, we have parallelized all major computations. These parallelizations are non-trivial and require low-level algorithmic optimizations, with challenges such as:

**Box-to-threadblock alignment** – A major challenge in the implementation of both the parallel far-field and near-field computations was correctly aligning thread blocks to boxes. This aligning requirement imposed non-trivial boundary conditions on data indexing when using shared memory. To minimize memory usage of box and block indicators for our implementation, we represented the box belonging of each point as index intervals (i.e. box 1 consists of points with $i \in [1, \ldots, 500]$ and box 2 with $i \in [501, \ldots, 1337]$, etc.) and modulo arithmetic to infer the block belonging. This clearly requires that the points are sorted or grouped according to their box belonging. However, as we detail in the next paragraph, arranging the points could not be done straightforwardly with native sorting methods. We further illustrate how parallelization is done for calculating near-field interactions in Appendix I.

**No native sorting methods** – We found that LibTorch [25] sorting methods often led to out-of-memory (OOM) due to allocation of large long-type vectors on GPU. When $n = 10^9$, this implies

allocating 8GB of memory, 25% of the 32GB card used, making it a necessity to avoid native sorting methods.

**In-place grouping data on boxes** – Due to infeasible LibTorch sorting methods, we additionally had to design an algorithm that finds a permutation that would group $\mathbf{X}$ into its corresponding boxes in linear time and memory. We used a count and increment-based strategy that would:

**1.** Count the number of points in each box during the assignment operation ($\mathcal{O}(n)$) and store the count in a vector $\xi$. Then run a cumulative sum over $\xi$, starting from 0.

**2.** Initialize a $n$ long permutation vector $\pi$. Us-

Input: A sorted matrix of interactions $I_0$

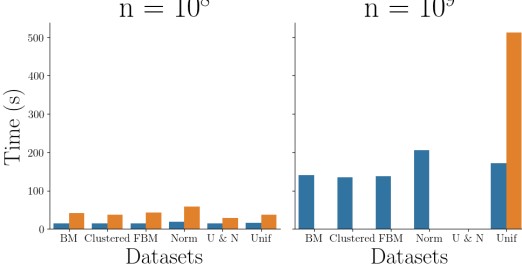

Figure 5: Here we assume $D = 1$, hence we only divide each box $2^D = 2$ times each time.

ing the counting vector $\xi$, we would re-run the assignment operation and arrange a point with index $i$ as following $\pi[\texttt{atomicAdd}(\xi'[\texttt{box\_index}], 1) + \xi[\texttt{box\_index}]] = i$, where $\xi'$ is a running count of points in each box. We specifically have to use the function `atomicAdd` to increment the count for each box in parallelized GPU environments to avoid thread locks.

We refer to the `box_division_cum_hash` and `box_division_assign_hash` function in `n_tree.cu` for exact details.

**Ensuring interactions are sorted** – To avoid any unnecessary sorting, we ensure that the matrix containing interactions

is always sorted by recursively dividing old interactions. We illustrate the procedure in Figure 5. We refer to the `get_new_interactions` function in `n_tree.cu` for the exact implementation.

However, we found that these optimizations and porting alone were not enough to scale to $n = 10^9$ on 3D datasets, as Figure 6 demonstrates. FFM doesn't remove empty boxes or handle boxes with few points in them and keeps exponentially creating new empty boxes and interactions, thus leading to out-of-memory (OOM) errors on non-uniform data (see Appendix 10).

Figure 6: KMVM times on several datasets between ■ FFM(GPU) vs ■ $\text{F}^3\text{M}$. U&N was only run up to $n = 5 \cdot 10^8$

## 4.2 Scaling to $n = 10^9$ on GPU ($\mathbf{F}^{2.5}\mathbf{M}$)

In this section, we detail the memory enhancements that allow $\mathrm{F}^3\mathrm{M}$ to consistently scale to $n = 10^9$.

**Removing empty boxes with hash list indexing**  To ensure linear memory on GPU, we only keep a reindexing vector $\sigma$ of size $n_x$(resp. $n_y$) in memory during the computation of the algorithm in addition to a list of interactions and box centers. This reindexing vector rearranges the data points so they appear in the order of the box they belong to. We optimize both the computation and the memory footprint of these objects by avoiding recursive formulas and hash lists.

Naively, points can be assigned to boxes by direct comparison to all existing box centers. As the number of centers grows exponentially with depth `tree_depth`, this method quickly becomes pathological. To amend this, we propose a linear complexity formula to retrieve the box index $\beta_i$ a

point $\mathbf{x} \in \mathbf{X} \subset \mathbb{R}^D$ belongs to $\beta_i = \underbrace{\sum_{d=1}^{D} 2^{\texttt{tree\_depth} \cdot (d-1)}}_{\text{Summing over } D \text{ dimensions}} \cdot \underbrace{\lfloor 2^{\texttt{tree\_depth}} \frac{x_d - \alpha_d}{\mathcal{E}} \rfloor}_{\substack{\in \{0,1\}, \text{ Denotes left or right} \\ \text{of center of box edge}}}$, where $\alpha_d$ denotes

the minimum value of $\mathbf{X}$ in dimension $D$ and $x_d$ is the value of $\mathbf{x}$ in dimension $D$. To prevent the number of boxes from growing exponentially, we remove empty boxes with each division. To assign points to the corresponding boxes, we use a hash list to store $\beta_i$ and the order $i$. We can then group points $\{\mathbf{x}_i\}_{i=1}$ to their respective ordering $i$ using the hash list in $\mathcal{O}(n)$ time in contrast to $\mathcal{O}(n \cdot 2^{D \cdot \texttt{tree\_depth}})$ by direct computation.

**Handling boxes with few points with *small field***  In cases when the number of points in each box can vary greatly, we separately consider the interactions where the number of points in boxes is small. Hence, we say that there is a *small field* interaction between boxes $B_p^X, B_q^X$ if both have a small number of points, i.e. if

$|B_p^X| + |B_q^X| \leq \rho$, for some threshold number $\rho$. To minimize the computations needed, $\rho$ can be set to $\rho = r_X + r_Y$. This intuitively allows $\mathrm{F}^3\mathrm{M}$ to directly compute interactions that are too small to benefit from interpolation savings (i.e. $|B_p^X| + |B_q^X| \leq r_X + r_Y$), thus limiting memory usage by stopping partitions from dividing further than necessary. In higher dimensions where the division rate is faster, $\rho$ can be set to a higher value to limit memory usage at the expense of more direct computations which are slower.

Figure 7: Full grid vs sparse grid.

**Sparse grids**  As the number of Lagrange polynomials increases exponentially with dimension, we implement sparse grids [29] to allow for a finer selection of interpolation nodes. With sparse grids, the number of nodes needed grows slower [20], thus saving memory. We give an example of a sparse grid versus a full grid in 2D in Figure 7.

## 4.3 Speeding up $\mathbf{F}^{2.5}\mathbf{M}$ $\left(\mathbf{F}^{2.5}\mathbf{M} \to \mathbf{F}^3\mathbf{M}\right)$

**Smoothness criteria**  FFM speeds up its computations with minimal loss in accuracy by selectively interpolating interactions that are far apart. To improve speed, we introduce the *smoothness criterion* to widen the selection of interactions that can be interpolated with minimal loss in accuracy. For a Gaussian Kernel $k(\mathbf{x}, \mathbf{y}) = \exp\left(\frac{\|\mathbf{x}-\mathbf{y}\|^2}{2\gamma^2}\right)$, with lengthscale $\gamma$, the smoothness criteria is defined as `is_smooth` $:= \frac{\frac{1}{N_{B_p^X}} \sum_{\mathbf{x}_i \in B_p^X} \sum_{d=1}^{D} \|x_i^{(d)} - \bar{x}^{(d)}\|^2}{2\gamma^2} + \frac{\frac{1}{N_{B_q^Y}} \sum_{\mathbf{y}_j \in B_q^Y} \sum_{d=1}^{D} \|y_j^{(d)} - \bar{y}^{(d)}\|^2}{2\gamma^2} \leq \eta$

between an adjacent interaction of boxes $B_p^X$, $B_q^Y$. The quantity computed can be understood as "Effective Variance" (EV), as it considers total variation in the exponent

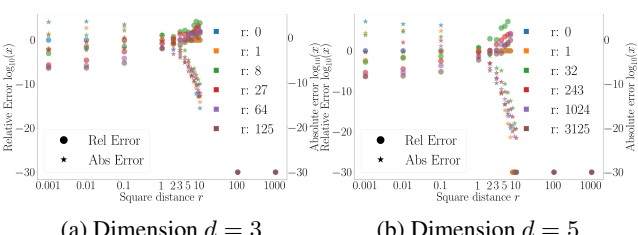

(a) Dimension $d = 3$  (b) Dimension $d = 5$

Figure 8: Plotting relative and absolute error against squared distance between boxes. "0" nodes mean we use the zero vector as an approximation to the KMVM for a gaussian kernel. We observe that its not beneficial to interpolate at all when the square distance exceeds 5 and that is is sufficient to only use $3^D$ nodes when square distance is $\leq 0.01$.

of the Gaussian kernel. We justify the *smoothness criteria* with the following proposition.

**Proposition 1.** *Consider* $\mathbf{x}, \mathbf{y} \in \mathcal{X} \subset \mathbb{R}^d$ *such that* $d(\mathbf{x}, \mathbf{y}) := \frac{\|\mathbf{x}-\mathbf{y}\|^2}{2\gamma^2} = \frac{1}{2\gamma^2} \sum_i^d (x^{(i)} - y^{(i)})^2 \leq \eta < 1$ *for all* $\mathbf{x}, \mathbf{y}$. *When interpolating* $k(\mathbf{x}, \mathbf{y}) = \exp\left(-d(\mathbf{x}, \mathbf{y})\right)$ *using bivariate Lagrange interpolation* $\mathcal{L}_r(\mathbf{x}, \mathbf{y}) := \mathbf{L}_X^T \cdot \mathbf{K} \cdot (\mathbf{L}_Y \cdot \mathbf{b})$ *with degree* $r = 2p$, *for any* $p \in \mathbb{N}_{>0}$ *there exist nodes* $\mathbf{s}^{\mathbf{x}}, \mathbf{s}^{\mathbf{y}}$ *for* $\mathcal{L}_r(\mathbf{x}, \mathbf{y})$ *such that the pointwise interpolation error is bounded by* $\mathcal{O}(\eta^{p+1})$.

See Appendix D for proof. Hence for small $\eta < 1$, we see that the error becomes small for well specified $\mathcal{L}_r(\mathbf{x}, \mathbf{y})$. To avoid calculating the sample variance during computations which costs $\mathcal{O}(n)$, we exploit that data is partitioned into hypercubes with a *known* edge $\mathcal{E}$ and take the upper bound of the variance in each cube as $\frac{\mathcal{E}^2}{4}$ along a dimension. A proof for this bound is provided in Appendix B. Adjacent interactions are then classified as smooth when $\sum_d^D \frac{\mathcal{E}^2}{4 \cdot 2\gamma}(\text{EV of } B_p^X) + \frac{\mathcal{E}^2}{4 \cdot 2\gamma}(\text{EV of } B_q^Y) = \frac{D\mathcal{E}^2}{\gamma^2 \cdot 4} \leq \eta$ which only costs $\mathcal{O}(1)$ to compute.

**Adaptive far-field approximation** To further improve speed we introduce an adaptive rule to select the number of interpolation nodes used when calculating far-field interactions. Error bounds for multidimensional Lagrange interpolation have been proposed in [22], however, these bounds cannot be directly used to create an adaptive interpolation rule. We thus simulate KMVM errors for $k(\mathbf{X}, \mathbf{Y}) \cdot \mathbf{b}$ where $\mathbf{X}, \mathbf{Y}$ are uniformly distributed and $\mathbf{b}$ is normally distributed. We fix a distance between $\mathbf{X}$ and $\mathbf{Y}$ and vary the squared of this distance between boxes against nodes in Figure 8. We use a Gaussian Kernel with $\gamma = \frac{1}{\sqrt{2}}$.

Based on Figure 8, we use the following rule for selecting the number of interpolation nodes for far-field interactions

$$
r_{\text{far}}(r) = \begin{cases} \min(r, 3^D) & \text{if } \left(\frac{\mathcal{E}}{2^{c_{\text{depth}}}}\right)^2 \cdot \frac{1}{2\gamma^2} \leq 0.01 \\ r & \text{if } 0.01 < \left(\frac{\mathcal{E}}{2^{c_{\text{depth}}}}\right)^2 \cdot \frac{1}{2\gamma^2} \leq 5 \\ 0 & \text{if } 5 < \left(\frac{\mathcal{E}}{2^{c_{\text{depth}}}}\right)^2 \cdot \frac{1}{2\gamma^2} \end{cases}
$$

where $r$ is the number of nodes chosen to interpolate with in the general case.

**Barycentric lagrange interpolation** We slightly improve the complexity further by implementing barycentric Lagrange interpolation [7] evaluated at the Chebyshev nodes of the second kind. As this is a well-known technique, we refer to the appendix for more details. It should be noted that the above methods can straightforwardly be extended to any *translation-invariant* kernel by recalculating the Taylor expansion for *smoothness criteria* and rerunning the simulation for *adaptive far-field approximation*.

### 4.4 Complexity

The time complexity of FFM is $\mathcal{O}(n \log(n))$ [3] and we use a similar derivation strategy for F³M to obtain a complexity that is dependent on the effective variance limit $\eta$ (chosen parameter) and the box width $\mathcal{E}$ (data). We first present two propositions needed to derive the complexity of F³M.

**Proposition 2.** *A far-field interaction between two boxes containing* $n_x$ *and* $n_y$ *points respectively has time complexity* $\mathcal{O}(n)$, *where* $n = \max(n_x, n_y)$.

**Proposition 3.** *Given* $n$ *data points in dimension* $D$, *the maximum number of divisions* $Tree_{\text{max divisions}}$ *is given by*

$$
Tree_{\text{max divisions}} = \log_{2^D}(n). \tag{2}
$$

With the above results, the complexity of FFM is taken as the maximum number of divisions multiplied by the complexity of far-field interactions at each division which yields $\mathcal{O}(n \log(n))$. We remark that near-field interactions between boxes containing only 1 data point have linear time complexity, hence the results hold.

**Theorem 1.** *Given a KMVM with edge* $\mathcal{E}$ *(dependent on data* $\mathcal{X}, \mathcal{Y}$*), lengthscale* $\gamma$, *effective variance limit* $\eta$, *n data points and data dimension* $D$, *F³M has time complexity* $\mathcal{O}\left(n \cdot \log_2\left(\frac{D \cdot \mathcal{E}^2}{\gamma^2 \cdot 4 \cdot \eta}\right)\right)$, *which can be taken as* $\mathcal{O}\left(n \cdot \log_2\left(\frac{C}{\eta}\right)\right)$ *where* $C \propto \frac{D \cdot \mathcal{E}^2}{\gamma^2}$.

**Memory footprint** As our implementation uses the same partitioning strategy as FFM, the *theoretical* memory complexity remains $\mathcal{O}(n)$ for F³M (see [3] for proof). However, this does not accurately reflect the memory footprint of the actual implementations, whose memory mostly depends

on the number of interactions stored. We summarize these memory footprints for FFM and F$^3$M in Theorem 2 below.

**Theorem 2.** *The number of interactions $M_i$ against tree depth $i$ of FFM and F$^3$M grows as* $\mathcal{O}\left(M_{i-1}2^{2\cdot D} - m_i^{far}\right)$ *and*

$$\mathcal{O}\left(M_{i-1}2^{2\cdot D} - (m_i^{empty})^2 - m_i^{far} - m_i^{smooth} - m_i^{small}\right)$$

*respectively. Here $M_{-1} = \frac{1}{2^{2D}}$ and $m_0^{far} = m_0^{smooth} = m_0^{small} = m_0^{empty} = 0$ and $m_i^{far}, m_i^{smooth}, m_i^{small}, m_i^{empty}$ denotes the number of far-field, smooth field, small field interactions and the number of empty boxes respectively at depth $i > 0$.*

We see that the additional approximations presented in F$^3$M also impacts memory footprint, as the additional $(m_i^{empty})^2, m_i^{smooth}, m_i^{small}$ terms removes a substantial amount of interactions at each $i$, significantly slowing down the growth of interactions, reducing memory growth. The efficacy of $(m_i^{empty})^2, m_i^{smooth}, m_i^{small}$ is widely dependent on data. As an example, data with points very close to each other would significantly benefit $m_i^{smooth}$ more, as the closeness of points would imply more smooth interactions. If points are sparsely spread out, $(m_i^{empty})^2, m_i^{small}$ would provide the most benefit as they remove empty boxes and stops boxes with few points to divide unnecessarily. All proofs can be found in Appendix E.

## 5 Experiments

We demonstrate the utility of F$^3$M over a variety of experiments using the Gaussian kernel $k(\mathbf{x}, \mathbf{y}) = \exp -(\frac{\|\mathbf{x}-\mathbf{y}\|^2}{2\gamma^2})$.[3] We generate data such that the EV (see section 4.3) varies between $0.1, 1, 10$ for data of sizes $n = 10^6, 10^7, 10^8, 10^9$. The parameters used for F$^3$M are $\eta = 0.1, 0.2, 0.3, 0.5$ and $r = 2^D, 3^D, 4^D$ with a cap at $r = 2048$. The error for the approximated KMVM product $\hat{\mathbf{v}}$ is calculated as Relative error $:= \frac{\|\hat{\mathbf{v}}-\mathbf{v}\|^2}{\|\mathbf{v}\|^2}$, where the true KMVM product $\mathbf{v}$ is obtained by calculating the full KMVM on a subset $\mathbf{X}'$ consisting of the first 5000 points in $\mathbf{X}$ against the entire dataset in double precision, i.e. $\mathbf{v} = k(\mathbf{X}', \mathbf{X}) \cdot \mathbf{b}$, where we fix $\mathbf{b} \sim \mathcal{N}(0, I_n)$. All experiments were run on NVIDIA V100-32GB cards, where the data is fitted entirely on the GPU. These cards were chosen since the extra graphic memory is necessary to fit the data on one card when $n = 10^9$. It should be noted that $n = 10^9$ can only be run up to $D = 3$, as $\mathbf{X}$ and $\mathbf{b}$ itself cannot fit in memory for higher dimensions with the GPUs we had available. For details on how F$^3$M scales across multiple GPUs, see Appendix G.

Table 2: F$^3$M (GPU) compared to results reported in [3] for FFM(CPU). F$^3$M achieves a $90\times$ speed up on a billion data points. F$^3$M used parameters $r = 64$ and $\eta = 0.5$

| | **FFM (12 CPU cores)** | | | **F$^3$M (GPU, Ours)** | | | |
| n | Time (s) | Error | Memory | Time (s) | Error | Memory | **Speedup** |
|---|---|---|---|---|---|---|---|
| $10^6$ | 33.4 | $1.35 \cdot 10^{-4}$ | 100 MB | $0.08 \pm 0.00$ | $3 \cdot 10^{-4} \pm 7 \cdot 10^{-5}$ | $\sim 28$ MB | $417\times$ |
| $10^7$ | 169 | $1.98 \cdot 10^{-4}$ | 1GB | $1.16 \pm 0.04$ | $3 \cdot 10^{-4} \pm 1.2 \cdot 10^{-4}$ | $\sim 280$ MB | $145\times$ |
| $10^8$ | 1499 | $1.81 \cdot 10^{-4}$ | 10 GB | $12.45 \pm 0.06$ | $2 \cdot 10^{-4} \pm 5 \cdot 10^{-5}$ | $\sim 2.8$ GB | $120\times$ |
| $10^9$ | 11340 | $3.11 \cdot 10^{-4}$ | 100 GB | $125.90 \pm 0.52$ | $3 \cdot 10^{-4} \pm 1.3 \cdot 10^{-5}$ | $\sim 28$ GB | $90\times$ |

**KMVM experiments** We consider a wide variation of generated datasets to simulate different real-world scenarios to test F$^3$M on. For the $k(\mathbf{X}, \mathbf{X})$-case we consider uniformly and normally distributed data ($D = 1, 2, 3, 4, 5, 6, 7$) together with data simulated from Brownian motion, fractional Brownian motion, and Clustered data ($D = 1, 2, 3$). For the $k(\mathbf{X}, \mathbf{Y})$-case we consider uniformly distributed $\mathbf{x}$ and normal distributed $\mathbf{y}$ ($D = 1, 2, 3, 4, 5, 6, 7$). See Appendix 10 for visualizations of data. We have to consider smaller $n$ for the $k(\mathbf{X}, \mathbf{Y})$-case when $D \geq 3$, as twice the amount of data needs to be stored. For $D = 3/(4, 5)/(6, 7)$ we instead consider at most $n = 5 \cdot 10^8/2.5 \cdot 10^8/10^8$. It should be noted that $D = 7$ is a hard limit for geometric partitioning-based methods, since for $D = 8$, we would have $2^{8 \cdot 2} \cdot 2^{8 \cdot 2} \approx 4.3 \cdot 10^9$ interactions after only 2 divisions. This number of interactions cannot even be represented by a 32-bit integer. We summarize the runs in Table 3 and plot the error and time complexity in Figure 9 for each dataset when $D = 3$. We find that F$^3$M maintains sub-linear empirical complexity up to $D = 6$, where we have to set *small field* limit $\rho$ to a larger number to

---

[3]F$^3$M is kernel agnostic, however we choose the Gaussian kernel for simplicity.

Table 3: Run time and relative error of all KMVM experiments for F³M. The slope is computed by regressing $\log_{10}(\text{Time (s)})$ against $\log_{10}(n)$. A slope of 1 implies $\mathcal{O}(n)$ scaling.

| $n/D$ | Time(s) | | | | | | | Relative Error | | | | | | |
|---|---|---|---|---|---|---|---|---|---|---|---|---|---|---|
| | 1 | 2 | 3 | 4 | 5 | 6 | 7 | 1 | 2 | 3 | 4 | 5 | 6 | 7 |
| $10^6$ | 0.2 ±0.1 | 0.2 ±0.2 | 0.2 ±0.1 | 0.4 ±0.2 | 0.9 ±0.9 | 3.0 ±2.4 | 3.1 ±2.1 | 0.0018 ±0.0021 | 0.0023 ±0.0028 | 0.0005 ±0.0009 | 0.0014 ±0.0013 | 0.0022 ±0.0022 | 0.0303 ±0.0314 | 0.0294 ±0.0282 |
| $10^7$ | 0.7 ±0.3 | 1.1 ±0.4 | 1.7 ±0.7 | 4.0 ±2.4 | 12.7 ±10.0 | 79.9 ±88.6 | 88.2 ±90.2 | 0.0016 ±0.0021 | 0.0019 ±0.0023 | 0.0006 ±0.0013 | 0.0017 ±0.0019 | 0.0057 ±0.0032 | 0.0325 ±0.0261 | 0.0273 ±0.0179 |
| $10^8$ | 3.5 ±1.1 | 7.4 ±2.1 | 17.1 ±7.3 | 40.6 ±28.7 | 76.0 ±83.2 | 525.1 ±410.9 | 512.9 ±471.4 | 0.002 ±0.0026 | 0.0025 ±0.0032 | 0.0007 ±0.0013 | 0.0023 ±0.0021 | 0.0051 ±0.0032 | 0.0376 ±0.0239 | 0.0444 ±0.017 |
| $2.5 \cdot 10^8$ | N/A | N/A | N/A | 76.7 ±25.9 | 340.7 ±247.6 | OOM | OOM | N/A | N/A | N/A | 0.0034 ±0.0029 | 0.0076 ±0.0012 | N/A | N/A |
| $5 \cdot 10^8$ | N/A | N/A | 74.9 ±28.5 | 289.4 ±222.8 | 631.5 ±997.0 | OOM | OOM | N/A | N/A | 0.0012 ±0.0013 | 0.0023 ±0.0026 | 0.0041 ±0.003 | N/A | N/A |
| $10^9$ | 29.4 ±8.9 | 73.8 ±23.4 | 174.0 ±76.6 | OOM | OOM | OOM | OOM | 0.0024 ±0.0024 | 0.0025 ±0.0027 | 0.0009 ±0.0018 | N/A | N/A | N/A | N/A |
| Slope | **0.78** | **0.85** | **0.99** | **0.98** | **0.99** | **1.06** | **1.02** | | | | | | | |
| $\mathcal{O}(n \log(n))$ | | | | 1.11 | | | | | | | | | | |

Table 4: FALKON using default KMVM vs FALKON with F³M.

| Dataset | $n$ | $D$ | $M$ | FALKON with default KMVM | | FALKON with F³M | | Error diff | Speedup |
|---|---|---|---|---|---|---|---|---|---|
| | | | | $R^2$ | Time (s) | $R^2$ | Time (s) | | |
| Uniform | $10^9$ | 3 | $10^5$ | $0.975 \pm 0.034$ | $7631 \pm 2$ | $0.976 \pm 0.038$ | $2234 \pm 429$ | 0% | 5.31 |
| Normal | $10^9$ | 3 | $10^5$ | $0.893 \pm 0.118$ | $7631 \pm 2$ | $0.902 \pm 0.114$ | $2234 \pm 429$ | 1% | 3.41 |
| OSM | $10^9$ | 2 | $10^5$ | $0.932 \pm 0.056$ | $6752 \pm 13$ | $0.943 \pm 0.043$ | $1670 \pm 48$ | 1.2% | 4.04 |
| NYC Taxi | $10^9$ | 3 | $10^5$ | $0.526 \pm 0.029$ (AUC) | $6963 \pm 69$ | $0.526 \pm 0.030$ (AUC) | $4535 \pm 7$ | 0% | 1.53 |

not run out of memory. Further, the error increases in the higher dimensions since we use fewer nodes per dimension when interpolating, owing to the *sparse grid* technique. We note that $D = 7$ has faster run times than $D = 6$ which is explained by that for some values of EV, $D = 7$ doesn't run with acceptable errors which skew the run time to datasets where a larger portion of the data can be interpolated.

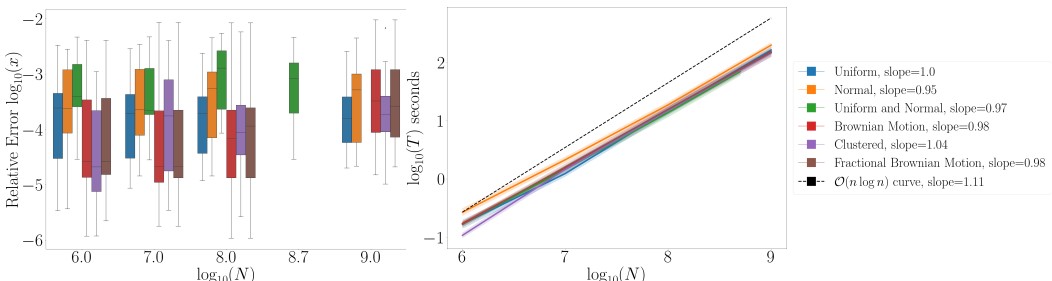

Figure 9: Relative error and time complexity for each 3D dataset.

We further replicate the data used in the first experiment in [3] and compare F³M against FFM (CPU) in Table 2.

**Kernel Ridge Regression experiment** We apply F³M to FALKON [24], where we replace their KMVM operation with F³M and compare performance and speed in solving Kernel Ridge Regression (KRR). The KMVM operation currently used for smaller dimensions is KeOps [11]. Given some data $\mathbf{X} \in \mathbb{R}^{N \times d}$ we want to find the solution $\alpha = (k(\mathbf{X}, \mathbf{X}) + \lambda I)^{-1} \mathbf{b}$ where $\lambda$ is the ridge parameter that stabilizes the inverse. FALKON is a Nyström approximation based solver that requires a subsample $\mathbf{X}' \in \mathbb{R}^{M \times d}$ of $\mathbf{X}$ to approximate the inverse computation. We focus the experiments on tall and skinny data and take $n = 10^9, d \leq 3$ with $M = 10^5$ for all experiments. We consider uniformly and normally sampled data, the Open Street Map (OSM) dataset [1] and a classification task on the NYC Taxi dataset [2], where we predict whether the customer will tip based on trip distance, trip time and fare cost. To construct $\mathbf{b}$ on synthetic problems, we first take a subset $\mathcal{D} \in \mathbb{R}^{1000 \times d}$ of $\mathbf{X}$ and sample $\alpha \sim \mathcal{N}(0, I_{1000 \times 1000})$. We then calculate $\mathbf{b} = k(\mathbf{X}, \mathcal{D}) \cdot \alpha + \varepsilon$, where $\varepsilon \sim \mathcal{N}(0, 0.1)$. We run KRR for $EV = 0.1, 1, 10$ on synthetic data, and report the average $R^2$ (AUC for NYC Taxi) and training time in Table 4. For the real world datasets OSM and NYC Taxi, we fix the lengthscale using the median heuristic proposed in [16] averaged our results over the 3 runs.

**Ablation study between FFM(GPU) and F³M** As much of the improved performance can be attributed to our GPU implementation, we conduct an ablation study of FFM(GPU) against F³M and KeOps in Table 5. We first present KMVM run times averaged over $D = 3$ and real-world datasets OSM and NYC Taxi. For KeOps, we only computed the KMVM on uniform data. Since KeOps is an exact method, the dataset distribution has no effect on computational time. Here, the *smoothness*

*criteria* and *adaptive far-field* technique improve computational time. We find that F³M achieves a speed-up between $2.0 - 33.3\times$ against FFM(GPU) and $8.0 - 8500\times$ speed-up against KeOps.

Table 5: Comparison between F³M, FFM(GPU) and KeOps. It should be noted that KeOps is only run up to $n = 10^8$ for all experiments (a run for $n = 10^9$ would take weeks). The times for $n > 10^8$ are extrapolated for KeOps. FFM(GPU) could only run on uniform data for $n = 10^9$.

| n | F³M time (s) | | | FFM(GPU) time (s) | | | KeOps time (s) | Speedup vs KeOps | Speedup vs (GPU) | | |
|---|---|---|---|---|---|---|---|---|---|---|---|
| | OSM $(D=2)$ | Taxi $(D=3)$ | $D=3$ | OSM $(D=2)$ | Taxi $(D=3)$ | $D=3$ | $D=3$ | | OSM $(D=2)$ | Taxi $(D=3)$ | $D=3$ |
| $10^6$ | $0.1\pm0.0$ | $0.5\pm0.2$ | $0.2\pm0.1$ | $0.7\pm0.2$ | $1.2\pm0.2$ | $0.4\pm0.2$ | 1.56 | 8.0 | **7.0** | **2.4** | **2.0** |
| $10^7$ | $1.1\pm0.4$ | $3.3\pm0.6$ | $1.7\pm0.7$ | $10.1\pm0.8$ | $110.0\pm1.9$ | $3.7\pm1.0$ | 148.6 | 87.0 | **9.2** | **33.3** | **2.2** |
| $10^8$ | $10.8\pm3.9$ | $30.4\pm6.9$ | $17.1\pm7.3$ | OOM | $769.5\pm27.9$ | $43.7\pm12.2$ | 1.49e4 | 873.0 | N/A | **25.3** | **2.6** |
| $10^9$ | $101.8\pm36.6$ | $290.0\pm69.4$ | $174.0\pm76.6$ | OOM | OOM | $517.4\pm60.8$ | 1.49e6 | 8583.0 | N/A | N/A | **3.0** |
| Error | 0.0005 | 0.0012 | 0.0008 | 0.0005 | 0.0002 | 0.0003 | 0.0 | | | | |
| Theoretical Complexity | $\mathcal{O}\left(n\cdot\log_2\left(\frac{D\cdot\mathcal{E}^2}{\gamma^2\cdot4\cdot\eta}\right)\right)$ | | | $\mathcal{O}(n\log(n))$ | | | $\mathcal{O}(n^2)$ | | | | |

**Ablation study between F²·⁵M and F³M** We provide and additional ablation study between F²·⁵M and F³M in Table 6. The results are quite similar to the comparison between F³M and FFM(GPU). Here, we see that smooth field and adaptive far-field approximation (F³M) both improve speed and also memory usage as smooth field helps approximate more interactions. We can thus infer empirically that the $m_i^{\text{smooth}}$ term in Theorem 2 has a significant impact on reducing the memory footprint of interactions.

Table 6: Comparison between F³M, F²·⁵M and KeOps. It should be noted that KeOps is only run up to $n = 10^8$ for all experiments (a run for $n = 10^9$ would take weeks). The times for $n > 10^8$ are extrapolated for KeOps.

| n | F³M time (s) | | | F²·⁵M time (s) | | | KeOps time (s) | Speedup vs KeOps | Speedup vs F²·⁵M | | |
|---|---|---|---|---|---|---|---|---|---|---|---|
| | OSM $(D=2)$ | Taxi $(D=3)$ | $D=3$ | OSM $(D=2)$ | Taxi $(D=3)$ | $D=3$ | $D=3$ | | OSM $(D=2)$ | Taxi $(D=3)$ | $D=3$ |
| $10^6$ | $0.1\pm0.0$ | $0.5\pm0.2$ | $0.2\pm0.1$ | $1.4\pm0.3$ | $2.0\pm0.3$ | $0.3\pm0.2$ | 1.56 | 8.0 | **14.0** | **4.0** | **1.5** |
| $10^7$ | $1.1\pm0.4$ | $3.3\pm0.6$ | $1.7\pm0.7$ | $11.3\pm0.4$ | $110.8\pm2.0$ | $4.5\pm2.1$ | 148.6 | 87.0 | **10.3** | **33.6** | **2.6** |
| $10^8$ | $10.8\pm3.9$ | $30.4\pm6.9$ | $17.1\pm7.3$ | OOM | $777.4\pm21.5$ | $42.5\pm13.1$ | 1.49e4 | 873.0 | N/A | **25.6** | **2.5** |
| $10^9$ | $101.8\pm36.6$ | $290.0\pm69.4$ | $174.0\pm76.6$ | OOM | OOM | $488.4\pm55.3$ | 1.49e6 | 8583.0 | N/A | N/A | **2.8** |
| Error | 0.0005 | 0.0012 | 0.0008 | 0.0004 | 0.0002 | 0.0016 | 0.0 | | | | |
| Theoretical Complexity | $\mathcal{O}\left(n\cdot\log_2\left(\frac{D\cdot\mathcal{E}^2}{\gamma^2\cdot4\cdot\eta}\right)\right)$ | | | $\mathcal{O}(n\log(n))$ | | | $\mathcal{O}(n^2)$ | | | | |

**Gaussian process regression experiments** We further compare F³M as a drop-in KMVM operation applied to Black-box Matrix Multiplication [15] for Gaussian Processes, compared to KISS-GP [35], an approximate Gaussian process using cubic interpolation for kernel approximation. We mimic the setup in [32] and consider the datasets 3DRoad, Song, Buzz and House Electric, where we apply PCA to the last three datasets and take the 3 first principal components for a fair comparison against KISS-GP, which is limited by $D \le 3$. We demonstrate the results in Table 7. As exact GP using F³M demonstrates competitive results even when compared to SVGP [18] and SGPR [30], we hypothesize that many high-dimensional datasets conform to the *manifold hypothesis* [13], allowing F³M to be widely applicable out-of-the-box even in high-dimensional settings.

Table 7: Gaussian process regression results. Exact GP using F³M shows improved results and scaling compared to KISS-GP. SGPR and KISS-GP could not scale to the HouseElectric dataset.

| Dataset | $n$ | $d$ | RMSE | | | | Training time (s) | | | |
|---|---|---|---|---|---|---|---|---|---|---|
| | | | Exact GP (F³M) | KISS-GP | SGPR $(m=512)$ | SVGP $(m=1024)$ | Exact GP (F³M) | KISS-GP | SGPR $(m=512)$ | SVGP $(m=1024)$ |
| 3DRoad | 278,319 | 3 | **0.297** $\pm0.036$ | 0.314 $\pm0.01$ | 0.661 $\pm0.010$ | 0.481 $\pm0.002$ | **27.8** $\pm$**18.0** | 312.9 $\pm10.8$ | 720.5 $\pm330.4$ | 2045.1 $\pm191.4$ |
| Song | 329,820 | 90 | **0.369** $\pm$**0.029** | 0.57 $\pm0.298$ | 0.803 $\pm0.002$ | 0.998 $\pm0.000$ | **7.2** $\pm$**3.1** | 1705.2 $\pm115.6$ | 2373.3 $\pm184.9$ | 473.3 $\pm187.5$ |
| Buzz | 373,280 | 77 | 0.967 $\pm0.002$ | 0.997 $\pm0.05$ | **0.300** $\pm$**0.004** | 0.304 $\pm0.012$ | **33.5** $\pm$**9.0** | 542.7 $\pm0.8$ | 1754.8 $\pm1099.6$ | 2780.8 $\pm175.6$ |
| HouseEletric | 1,311,539 | 9 | 0.308 $\pm0.006$ | OOM | OOM | **0.084** $\pm$**0.005** | **79.8** $\pm$**23.1** | N/A | N/A | 22062.6 $\pm282.0$ |

# 6   Limitations and Further Research

This work has introduced and implemented F³M on GPU, which enables fast KMVM for tall and skinny data up to $n = 10^9$. F³M has improved complexity which also is controllable through $\eta$, and retains linear memory. Experiments in higher dimensions also exhibit linear complexity, however requiring more nodes for lower errors. F³M can further be directly used as a drop-in KMVM operation, as demonstrated with FALKON and Gaussian process regression, achieving significant speedups and competitive performance on both tasks. As an interpolation based approximation method, F³M is still limited by the exponential growth of interpolation nodes with respect to $D$, although removing empty boxes, *small field* and sparse grids allow KMVM for $D \le 7$. A fruitful direction would be to extend ideas in F³M to accommodate higher-dimensional data by considering randomized partitioning [4], decoupling the dependency on $D$ in geometry based partitioning. Further, an exact characterization of how $m_i^{\text{far}}, m_i^{\text{smooth}}, m_i^{\text{small}}, m_i^{\text{empty}}$ grows is left to future work.

## Acknowledgments

The authors sincerely thank Lood van Niekerk and Jean-François Ton for their helpful comments.

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
