

| (a) Uniform dataset | (b) Normal dataset | (c) Uniform and Normal dataset |
| (d) Clustered dataset | (e) Fractional Brownian Motion | (f) Brownian motion |

Figure 10: 2D illustrations of the synthetic datasets.

## A  Synthetic data

**Note on synthetic datasets**    We generated synthetic datasets of different types to measure the ability of $F^3M$ to deal with dense or sparse data. Dense datasets were generated as independent samples with either uniform or normal distributions. Clustered datasets were generated by sampling cluster centers from a normal distribution, and then recursively sampling sub-cluster centers from a normal distribution with reduced standard deviation and centered at each cluster center, until the desired number of points is attained. Fractional Brownian Motion and Brownian Motion samples were generated as samplings of Fractional Brownian Motion paths with respective Hurst index 0.75 and 0.5. Figure 10 shows samples of each dataset type in the 2D case.

## B  Note on maximal variance on an interval

**Proposition 4.** *Consider a random variable $X \in \mathbb{R}$ with finite variance with $m = \inf X$ and $M = \sup X$. Then $Var(X) \leq \frac{(M-m)^2}{4}$.*

*Proof.* Define a function $g$ by $g(t) = \mathbb{E}\left[(X - t)^2\right]$. Computing the derivative $g'$, and solving $g'(t) = -2\mathbb{E}[X] + 2t = 0$ yields that $g$ achieves its minimum at $t = \mathbb{E}[X]$ (note that $g'' > 0$). Now, consider the value of the function $g$ at the special point $t = \frac{M+m}{2}$. It must be the case that $\text{Var}[X] = g(\mathbb{E}[X]) \leq g\left(\frac{M+m}{2}\right)$. Evaluating yields the expression

$$g\left(\frac{M + m}{2}\right) =$$

$$\mathbb{E}\left[\left(X - \frac{M + m}{2}\right)^2\right] = \frac{1}{4}\mathbb{E}\left[((X - m) + (X - M))^2\right]$$

Since $X - m \geq 0$ and $X - M \leq 0$, we have

$$((X - m) + (X - M))^2 \leq ((X - m) - (X - M))^2 = (M - m)^2$$

implying that

$$\frac{1}{4}\mathbb{E}\left[((X - m) + (X - M))^2\right] \leq$$

$$\frac{1}{4}\mathbb{E}\left[((X - m) - (X - M))^2\right] = \frac{(M - m)^2}{4}$$

Hence

$$\text{Var}[X] \leq \frac{(M-m)^2}{4}$$

$\square$

## C  Details on barycentric Lagrange interpolation

The barycentric lagrange interpolation is written as

$$L_i(t) = \frac{\frac{w_i}{t-s_i}}{\sum_{i=0}^{r} \frac{w_i}{t-s_i}}, w_i = \frac{1}{\prod_{j=0, j\neq i}^{r}(s_i - s_j)}, i = 0, \ldots, n$$

where $w_i$ are known as the barycentric weights. In case of singularities, i.e. when $t = s_j$, we set $L_i(s_j) = \delta_{ij}$. In particular, [7] proposes Chebyshev nodes of the second kind $s_i = \cos\theta_i$, $\theta_i = \frac{i\pi}{r}$, $i = 0, \ldots, r$. This choice of nodes combined with the scale invariance property of the barycentric form makes the calculation of $w_i$ particularly easy

$$w_i = (-1)^i \delta_i, \quad \delta_i = \begin{cases} 1/2, & i = 0 \text{ or } i = r \\ 1, & i = 1, \ldots, r-1 \end{cases}$$

and reduces the complexity of calculating $w_i$ from $\mathcal{O}(r^2)$ to $\mathcal{O}(r)$. The Lagrange interpolation polynomial can then be expressed as $p_r(t) = \sum_{i=0}^{r} \frac{\frac{w_i}{t-s_i}}{\sum_{i=0}^{r} \frac{w_i}{t-s_i}} f_i$.

## D  Smooth field proof

**Proposition.** *Consider* $\mathbf{x}, \mathbf{y} \in \mathcal{X} \subset \mathbb{R}^d$ *such that* $d(\mathbf{x}, \mathbf{y}) := \frac{\|\mathbf{x}-\mathbf{y}\|^2}{2\gamma^2} = \frac{1}{2\gamma^2}\sum_i^d (x^{(i)} - y^{(i)})^2 \leq \eta < 1$ *for all* $\mathbf{x}, \mathbf{y}$. *When interpolating* $k(\mathbf{x}, \mathbf{y}) = \exp(-d(\mathbf{x}, \mathbf{y}))$ *using bivariate Lagrange interpolation* $\mathcal{L}_r(\mathbf{x}, \mathbf{y}) := \mathbf{L}_X^T \cdot \mathbf{K} \cdot (\mathbf{L}_Y \cdot \mathbf{b})$ *with degree* $r = 2p$, *for any* $p \in \mathbb{N}_{>0}$ *there exist nodes* $\mathbf{s}^\mathbf{x}, \mathbf{s}^\mathbf{y}$ *for* $\mathcal{L}_r(\mathbf{x}, \mathbf{y})$ *such that the pointwise interpolation error is bounded by* $\mathcal{O}(\eta^{p+1})$.

*Proof.* Note that $k(\mathbf{x}, \mathbf{y})$ is analytic in $d(\mathbf{x}, \mathbf{y})$ with Taylor expansion given by $T_p e^{-d(\mathbf{x}, \mathbf{y})} = 1 - d(\mathbf{x}, \mathbf{y}) + \ldots + \mathcal{O}(d(\mathbf{x}, \mathbf{y})^{p+1})$. With $d(\mathbf{x}, \mathbf{y}) \leq \eta$, it follows that $|T_p k(\mathbf{x}, \mathbf{y}) - k(\mathbf{x}, \mathbf{y})| \leq \mathcal{O}(\eta^{p+1})$. Using triangle inequality, we have $|\mathcal{L}_r(\mathbf{x}, \mathbf{y}) - k(\mathbf{x}, \mathbf{y})| \leq |T_p k(\mathbf{x}, \mathbf{y}) - k(\mathbf{x}, \mathbf{y})| + |T_p k(\mathbf{x}, \mathbf{y}) - \mathcal{L}_r(\mathbf{x}, \mathbf{y})| \leq \eta^{p+1} + |T_p k(\mathbf{x}, \mathbf{y}) - \mathcal{L}_r(\mathbf{x}, \mathbf{y})|$. We note that $\mathcal{L}_r(\mathbf{x}, \mathbf{y})$ contains all the terms of the Taylor expansion, and we can thus choose the nodes $\mathbf{s}^\mathbf{x}, \mathbf{s}^\mathbf{y}$ of $\mathcal{L}_r(\mathbf{x}, \mathbf{y})$ such that $|T_p k(\mathbf{x}, \mathbf{y}) - \mathcal{L}_r(\mathbf{x}, \mathbf{y})| = 0$ as long as $r = 2p$, meaning the polynomial orders are matched. $\square$

*Note that the same proof strategy can be applied to any kernel $k$ that admits a Taylor expansion.*

## E  Complexity

**Proposition.** *A far-field interaction between two boxes containing $n_x$ and $n_y$ points respectively has time complexity $\mathcal{O}(n)$, where $n = \max(n_x, n_y)$.*

*Proof.* Far-field interactions are calculated as

$$\mathbf{v} \approx \underbrace{\mathbf{L}_X^T \cdot}_{\mathcal{O}(n_x \cdot r_X)} (\underbrace{\mathbf{K} \cdot}_{\mathcal{O}(r_X \cdot r_Y)} \underbrace{(\mathbf{L}_Y \cdot \mathbf{b})}_{\mathcal{O}(n_y \cdot r_Y)}). \tag{3}$$

As $r_X, r_Y$ are independent of $n_x, n_y$, the complexity becomes $\mathcal{O}(n)$. Further see [3] for alternative proof. $\square$

**Proposition.** *Given $n$ data points in dimension $D$, the maximum number of divisions $Tree_{max\ divisions}$ is given by*

$$Tree_{max\ divisions} = \log_{2^D}(n). \tag{4}$$

*Proof.* To see this, simply solve for

$$\frac{n}{2^{D \cdot \text{Tree}_{\texttt{max divisions}}}} = 1 \implies \text{Tree}_{\texttt{max divisions}} = \log_{2^D}(n).$$

Further see [3] for alternative proof. □

**Theorem.** *Given a KMVM with edge $\mathcal{E}$ (dependent on data $\mathcal{X}, \mathcal{Y}$), lengthscale $\gamma$, effective variance limit $\eta$, $n$ data points and data dimension $D$, $F^3M$ has time complexity $\mathcal{O}(n \cdot \log_2 \left( \frac{D \cdot \mathcal{E}^2}{\gamma^2 \cdot 4 \cdot \eta} \right))$, which can be taken as $\mathcal{O}(n \cdot \log_2 \left( \frac{C}{\eta} \right))$ where $C \propto \frac{D \cdot \mathcal{E}^2}{\gamma^2}$.*

*Proof.* Recall that near-field interactions can be smoothly interpolated when $D \cdot \frac{\mathcal{E}^2}{\gamma^2 \cdot 4 \cdot 2^{\texttt{tree\_depth}}} \leq \eta$. Then all interactions will be interpolated when $\texttt{tree\_depth} \geq \log_2 \left( \frac{D \cdot \mathcal{E}^2}{\gamma^2 \cdot 4 \cdot \eta} \right)$, which implies we can take $\text{Tree}_{\text{max divisions}} = \log_2 \left( \frac{D \cdot \mathcal{E}^2}{\gamma^2 \cdot 4 \cdot \eta} \right)$. Hence the complexity is $\mathcal{O}(n \cdot \log_2 \left( \frac{D \cdot \mathcal{E}^2}{\gamma^2 \cdot 4 \cdot \eta} \right))$. □

**Theorem.** *The number of interactions $M_i$ against tree depth $i$ of FFM and $F^3M$ grows as $\mathcal{O}\left( M_{i-1} 2^{2 \cdot D} - m_i^{far} \right)$ and*

$$\mathcal{O}\left( M_{i-1} 2^{2 \cdot D} - (m_i^{empty})^2 - m_i^{far} - m_i^{smooth} - m_i^{small} \right)$$

*respectively. Here $M_{-1} = \frac{1}{2^{2D}}$ and $m_0^{far} = m_0^{smooth} = m_0^{small} = m_0^{empty} = 0$ and $m_i^{far}, m_i^{smooth}, m_i^{small}, m_i^{empty}$ denotes the number of far-field, smooth field, small field interactions and the number of empty boxes respectively at depth $i > 0$. Note that for $i > 0$, these are dependent on data.*

*Proof.* We prove through induction that the recursion holds for $F^3M$. We start with the base case $M_0 = 1$, since at depth $i = 0$, we only have one box and hence only one interaction. $M_0 = M_{-1} 2^{2 \cdot D} - (m_0^{empty})^2 - m_0^{far} - m_0^{smooth} - m_0^{small} = 1$. Clearly at depth 0, there can not be any empty boxes or possible approximations. For the induction step, $M_i = M_{i-1} 2^{2 \cdot D} - (m_i^{empty})^2 - m_i^{far} - m_i^{smooth} - m_i^{small}$. To get $M_{i+1}$, each box at depth $i$ is first divided into $2^D$, hence the number of interactions grows by $2^{2 \cdot D}$. At depth $i + 1$, we can further remove $(m_{i+1}^{empty})^2$ interactions between empty boxes and further compute $m_{i+1}^{far} + m_{i+1}^{smooth} + m_{i+1}^{small}$ interactions. Then $M_{i+1} = (M_{i-1} 2^{2 \cdot D} - (m_i^{empty})^2 - m_i^{far} - m_i^{smooth} - m_i^{small}) 2^{2 \cdot D} - (m_{i+1}^{empty})^2 - m_{i+1}^{far} - m_{i+1}^{smooth} - m_{i+1}^{small} = M_i 2^{2 \cdot D} - (m_{i+1}^{empty})^2 - m_{i+1}^{far} - m_{i+1}^{smooth} - m_{i+1}^{small}$. Thus the base case and induction step holds which completes our proof. This proof also covers FFM, since FFM can be as a special case for $F^3M$ without removing empty boxes, smooth field and small field computation. □

# F   Algorithm summary

We present a summary of the algorithm presented in FFM in Algorithm 1 and the **modifications $F^3M$ does in boldface**.

**Algorithm 1:** FFM ($\mathbf{F}^3\mathbf{M}$)

---

**Input:** Datasets $\mathbf{X}, \mathbf{Y}, \mathbf{b}$, kernel $k$, average points threshold $\zeta$
**Result:** $\mathbf{v} = k(\mathbf{X}, \mathbf{Y}) \cdot \mathbf{b}$
Initialize treecodes $\tau_x = T(\mathbf{X}), \quad \tau_y = T(\mathbf{Y})$
Initialize near-field interactions as $\mathcal{I}_{\text{near}} = \{0, 0\}$
Initialize output $\mathbf{v} = \mathbf{0}$
**while** $|\mathcal{I}| > 0$ *and* $\mathtt{MaximumBoxSize}(\tau_x) > \zeta$ *and* $\mathtt{MaximumBoxSize}(\tau_y) > \zeta$ **do**
    Divide $\tau_x, \tau_y$
    Calculate interactions left $\mathcal{I} := f(\mathcal{I}_{\text{near}})$
    Partition $\mathcal{I}$ to $\{\mathcal{I}_{\text{near}}, \mathcal{I}_{\text{far}}, \boldsymbol{\mathcal{I}}_{\text{smooth}}\}$
    **Throw away interactions that are too far from eachother**
    Compute far-field interactions $\mathbf{v} \mathrel{+}= \text{FarFieldCompute}(\tau_x, \tau_y, \mathcal{I}_{\text{far}})$
    **Compute smooth field interactions $\mathbf{v} \mathrel{+}= \mathbf{FarFieldCompute}(\tau_x, \tau_y, \boldsymbol{\mathcal{I}}_{\text{smooth}})$** ;
**end**
Compute remaining near-field interactions $\mathbf{v} \mathrel{+}= \text{NearFieldCompute}(\tau_x, \tau_y, \mathcal{I}_{\text{near}})$

---

Instead of comparing the average box size to $\zeta$ we compare the maximum box size. When points are non-uniformly distributed, taking the maximum ensures that we don't compute near-field interactions on boxes with many points, since it will be inefficient.

## G  Scalability Analysis

We conduct a scalability analysis over $N_{\text{GPU}} = 1, 2, 4, 8$. We parallelize the KMVM product by considering the $k(\mathbf{X}, \mathbf{X})$-case and divide the work onto multiple GPUs by partitioning each subproduct of the KMVM (see for Figure 11 an example when $N_{\text{GPU}} = 8$). We take $\mathbf{X}$ to be Uniform and 3 dimensional. We present results in Figure 12.

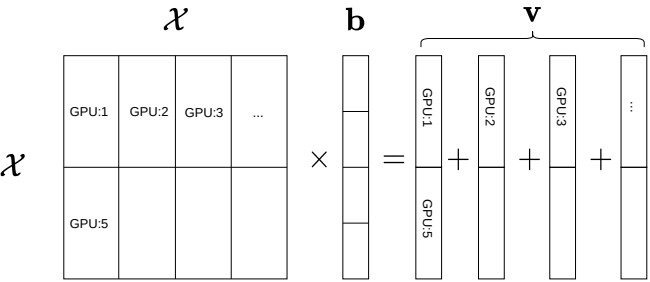

Figure 11: Partitioning a KMVM product into 8 jobs

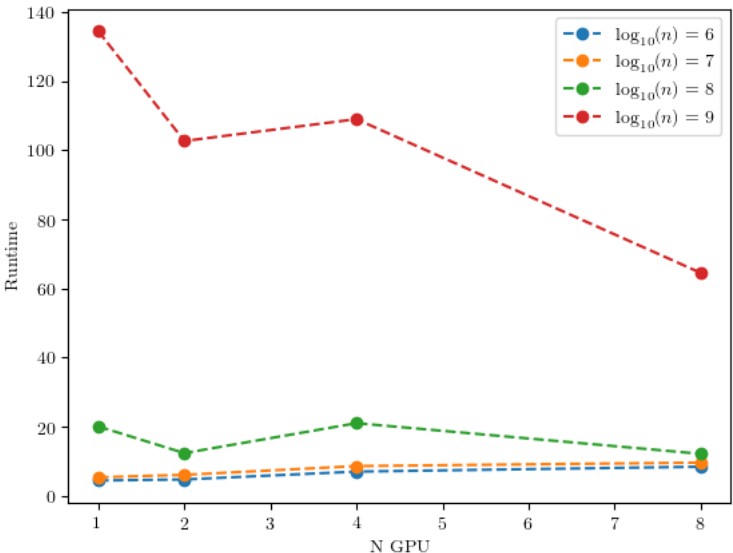

Figure 12: Since the V100 cards we use have very high throughput, we only get a performance boost when $n = 10^9$.

We also use `nvprof` to analyze the % of peak throughput of the V100 cards F$^3$M can utilize. We run `nvprof` for 3 dimensional uniform data for $n = 10^6, 10^7, 10^8, 10^9$. We present our results in Figure 13.

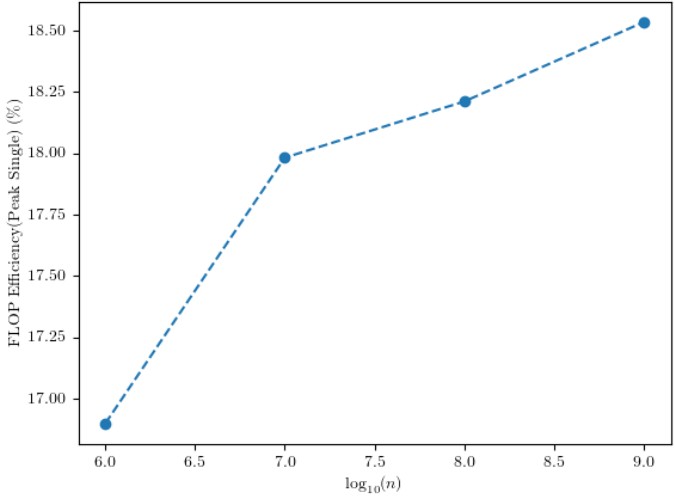

Figure 13: We used the `flop_sp_efficiency` metric in `nvprof` to generate the plot. One GPU was used for this experiment.

# H   Impact of $\eta$ and $r$ on performance

The performance of F$^3$M is tuned by choosing $\eta$ and $r$ to trade speed against accuracy. In Figure 14 we plot how different choices of $\eta$ and $r$ impacts computation time for F$^3$M on 3D data.

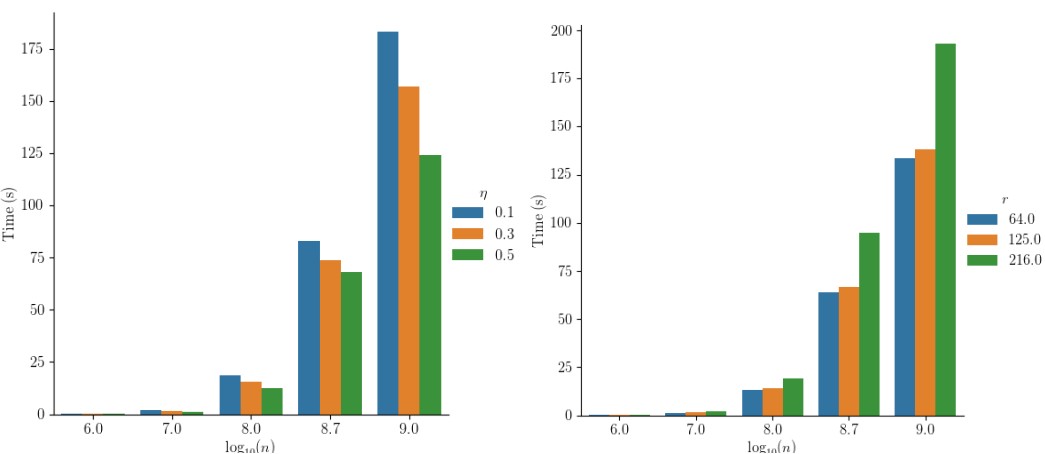

Figure 14: We see that larger $\eta$, more aggressive `smoothness criteria` and smaller number of interpolation nodes $r$ improve speed.

# I  Implementation overview

We provide a skiss of how data is stored and used for $F^3M$ in Figure 15.

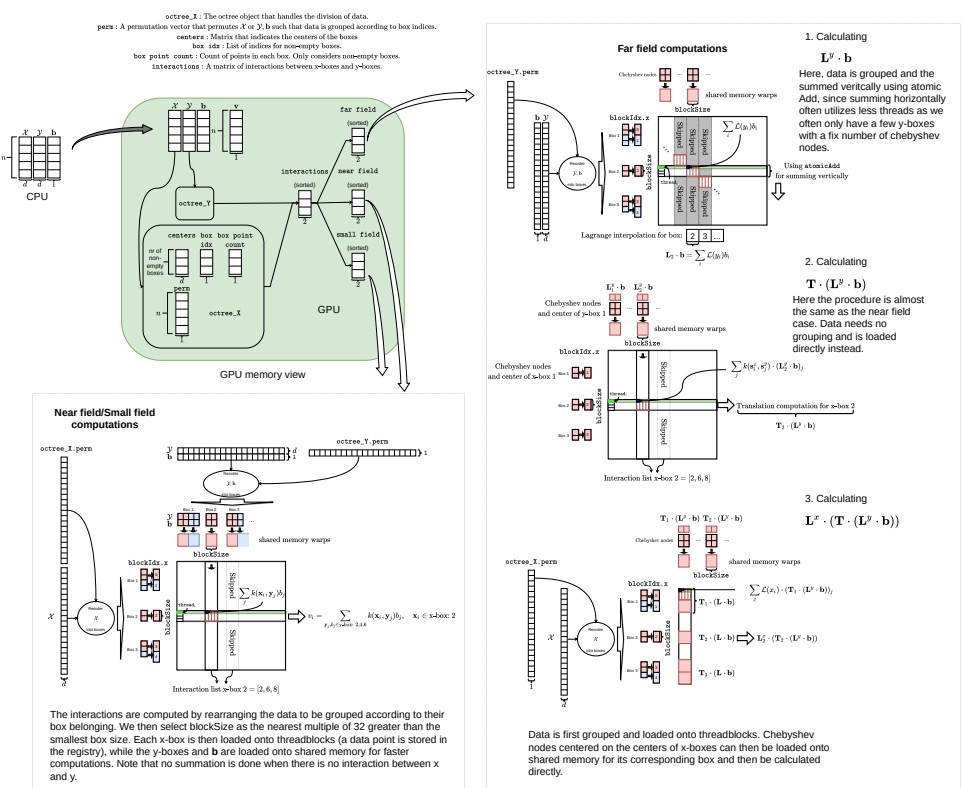

Figure 15: Skiss of how data is stored and used on GPU.

We provide an illustration on near field computations are carried out for $F^3M$ in Figure 16.

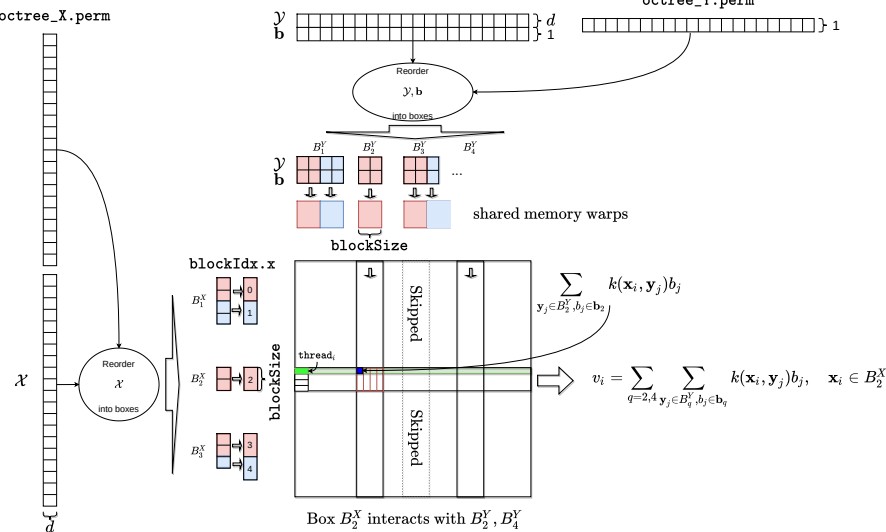

Figure 16: Illustration of how near-field interactions are computed in parallel on GPU. First data is reordered into their corresponding boxes using a permutation vector. Then each box is loaded in parallel into several thread blocks (indexed by `blockIdx.x`), wherein the challenge lies to execute this correctly. The computations are then parallelized across blocks, where only interactions are computed.