# OpenReview forum: "Giga-scale Kernel Matrix-Vector Multiplication on GPU"
_NeurIPS.cc/2022/Conference — NeurIPS 2022 Accept_

### Official Review · Reviewer_eghV · 2022-07-10

**Rating:** 6
**Confidence:** 2
**Soundness:** 3 good
**Presentation:** 3 good
**Contribution:** 3 good

**Summary:**

This paper tries to accelerate the fundamental problem of kernel matrix-vector multiplication (KMVM) operation. The authors propose F^{2.5}M and which is built on classical FFM and specially designed for GPU parallelization. Then they introduce the main algorithm F^{3}M to further accelerate the computation speed. The experiments show that compared with CPU and GPU baselines the new proposed algorithm uses less memory and has a faster speed while controlling the error in an acceptable range.

**Questions:**

I would like to to see some ablation studies and experiments on F^{2.5}M.

**Strengths And Weaknesses:**

This paper tries to solve the fundamental problem of high memory usage and low speed in large-scale KMVM operations. The writing is clear and the motivation is well-driven. The algorithm is straightforward but seems effective according to the experiment results. The figures in the attachment materials show clearly how the algorithm utilizes the GPU parallelization attribute.
Minor:
Pictures need to be replaced with vector graphics to improve clarity.

---

> ### Author Response · Authors · 2022-07-31
> **Response**
>
> Thank you for your time and effort in reviewing the paper! We respond to your comments and questions below:
>
> *Q1*: I would like to to see some ablation studies and experiments on F^{2.5}M.
> *A1*: We have provided the requested experiments with commentary in Appendix J (last page), Table 7 of the rebuttal revision.
>
> We will update the figures with vector graphics, thank you for the suggestion!

---

### Official Review · Reviewer_sNpd · 2022-07-11

**Rating:** 6
**Confidence:** 2
**Soundness:** 3 good
**Presentation:** 3 good
**Contribution:** 3 good

**Summary:**

This paper presents a fast matrix vector multiplication for skinny matrices.


**Questions:**

1. Is it possible to use tensor cores in the proposed algorithm?
2. How to improve the FLOP efficiency?

**Limitations:**

As the authors claim, the major limitation of the proposed algorithm is the restriction of the dimension (D <= 7).
Although it may limit the usage, many relational tabular data fit the constraint.

**Strengths And Weaknesses:**

Strong Points
----
1. This paper targets to solve a well-motivated and important problem.
2. Related work is clearly discussed.
3. The empirical results substantially outperform baselines.

Weak Points
---
1. The V100 used in the experiments supports tensor cores. I suggest the authors include a discussion of whether tensor cores may be utilized in F^3M.
2. The GPU utilization of the proposed algorithm is relatively low to the full capacity of V100. I wonder whether the authors could add a discussion of the reason and how to improve the utilization.

Presentation
----
The upper part of Figure 5 is incomplete. And the text is too small for readers.

---

> ### Author Response · Authors · 2022-07-31
> **Response**
>
> Thank you for your time and effort in reviewing the paper! We respond to your comments and questions below:
>
> *Q1*: Is it possible to use tensor cores in the proposed algorithm?
> *A1*: We have not attempted to utilize tensor cores in the proposed algorithm yet, and we believe some parts of the code benefit from tensor cores. However as tensor cores mostly require floating point 16 accuracy, one must be careful about precision when attempting this. We view this as a very promising venue for further development in scaling further.
>
> *Q2*: How to improve the FLOP efficiency?
> *A2*: That's a good question! Through our profiling, we found that a potential bottleneck currently is when the Lagrange interpolation occurs between each interaction, as it is quite difficult to parallelize very effectively. These circumstances can directly be attributed to each box having a different amount of points. Another aspect is that quite a lot of memory is reallocated during the run, which hampers throughput. We plan to detail this carefully in a more comprehensive fashion in a future revision.

---

### Official Review · Reviewer_MjYH · 2022-07-11

**Rating:** 6
**Confidence:** 3
**Soundness:** 2 fair
**Presentation:** 1 poor
**Contribution:** 3 good

**Summary:**

The paper considers speeding up the matrix-vector multiplication operation in kernel methods for a specific structure (tall and skin) of the kernel matrix. The approximation method builds on the FFM algorithm, which separates data points into far-field and near-field interactions and Lagrange interpolation. The proposed schemes are further implemented on GPUs to exploit their parallel processing power; various speed-ups can be observed compared to the FFM method.

**Questions:**

The last two items in Table 1 are quite strange as runtime depends on the hardware. What does that mean by running under a minute or an hour?

It seems that the F3 method is specific to the Gaussian kernel in Section 4.3? It is not clear how the methods depend on hyperparameters when the data set is changed. For example, the suggested parameters in Line 250 are based solely on Figure 7, which may not generalize to new data sets.

Theorem 2 is hard to understand, it needs more discussion. It is not clear how large the different terms (with negative signs) are, relative to the first term in the complexity bound.

**Limitations:**

see above

**Strengths And Weaknesses:**

The methods contain several levels of approximation and heuristics, with many hyperparameters to be tuned. It is not clear how sensitive these methods are to these parameters. How to best select them for a given dataset?

The paper is densely packed with approximations and notations, which make it quite difficult to follow at some points.
In Section 4.1, the authors repeatedly mentioned **we used non-trivial method...** without any guidance. Please consider updating it. Some parameters are used for a different meaning, e.g., the vector **v** in Line 143 and Line 144.

---

> ### Author Response · Authors · 2022-07-31
> **Response**
>
> Thank you for your time and effort in reviewing the paper! We respond to your comments and questions below:
>
> *Q1*: The methods contain several levels of approximation and heuristics, with many hyperparameters to be tuned. It is not clear how sensitive these methods are to these parameters. How to best select them for a given dataset?
> *A1*: We give some guidance on the impact of hyperparameters effective variance limit $\eta$ and the number of interpolation nodes $r$ in Figure 13 in appendix H. By increasing the number of interpolation nodes (increasing accuracy) the computation is slower while increasing effective variance limit (more interactions interpolated) the performance improves. There is a controllable trade-off between accuracy and performance when tuning $\eta$ and $r$.
>
>
> *C2*: The paper is densely packed with approximations and notations, which make it quite difficult to follow at some points. In Section 4.1, the authors repeatedly mentioned we used non-trivial method... without any guidance. Please consider updating it. Some parameters are used for a different meaning, e.g., the vector v in Line 143 and Line 144.
> *A2*: Thank you for the suggestion! We will extend the paragraphs about low-level implementations by including more pseudo-code and more details about the algorithmic challenges encountered during this project. As these changes require more careful writing, we have decided to not include them in the rebuttal revision in hopes of providing a more comprehensive overview. We have clarified the notation for vector $v$ in lines 143 and 144 in the rebuttal revision.
>
> *Q3*: The last two items in Table 1 are quite strange as runtime depends on the hardware. What does that mean by running under a minute or an hour?
> *A3*: It means we have run FFM(GPU) and F$^3$M on V100 chips on 1 billion datapoints on 3D data. We have added a footnote in the rebuttal revision to clarify this. Thanks for pointing this out!
>
> *C4*: It seems that the F3 method is specific to the Gaussian kernel in Section 4.3? It is not clear how the methods depend on hyperparameters when the data set is changed. For example, the suggested parameters in Line 250 are based solely on Figure 7, which may not generalize to new data sets.
> *A4*: The Gaussian kernel is used as a proof of concept for the method, the method and derivations can be extended to other kernels either directly or with minor modification depending on the kernel. One can quite easily rerun the procedure in Figure 7 for a dataset to validate parameter selection. However, the datasets and selection we propose should generally work well. It is however an important point we hope future work can resolve.
>
> *Q5*: Theorem 2 is hard to understand, it needs more discussion. It is not clear how large the different terms (with negative signs) are, relative to the first term in the complexity bound.
> *A5*: We agree. We have reformulated and clarified the meaning of the theorem in the rebuttal revision. New content marked in blue.

---

### Official Review · Reviewer_TBZE · 2022-07-12

**Rating:** 7
**Confidence:** 4
**Soundness:** 4 excellent
**Presentation:** 3 good
**Contribution:** 3 good

**Summary:**

The paper proposes a novel algorithm to approximate kernel matrix-vector multiplication (KMVM), for large kernels and data of dimension less than or equal to 7. Experiments and theoretical analysis suggest the algorithm has linear time and memory complexity. Since KMVM is a key component of other algorithms like conjugate gradients, the authors show their algorithm can be combined with existing large-scale kernel methods and demonstrate significant speed ups.

**Questions:**

- Maybe mention that $D$ is the dimension of the data in the first paragraph where it is introduced.
- Are there any assumptions on the type of kernel F$^3$M is useful for? It seems like most experiments were performed on RBF kernels.
- Line 143 should $u_1$ and $u_2$ be $v_1$ and $v_2$?
- Can the authors include more detail in Section 4.1 on the low-level GPU optimizations performed in the main text?

**Limitations:**

No concerns about negative societal impact.

**Strengths And Weaknesses:**

Strengths
- The limitations of the method are clearly discussed.
- The paper makes a strong contribution with a nice algorithmic advance and a technically challenging implementation on GPU.
- All the main claims of the paper are thoroughly supported by experiments, including an ablation study. Overall a high quality paper.

Weaknesses
- The algorithm is limited to low-dimensional data, which might hamper the paper's significance in the NeurIPS community. However, this limitation is clearly acknowledged and discussed.
- I found some sections of the paper hard to follow, particularly Section 4.2, but this may be because I'm not familiar with some of the prior work the paper builds on.

---

> ### Author Response · Authors · 2022-07-31
> **Response**
>
> Thank you for your time and effort in reviewing the paper! We respond to your comments and questions below:
>
> *Q1*: Maybe mention that $D$ is the dimension of the data in the first paragraph where it is introduced.
> *A1*: Thank you for this suggestion! We have done it in an uploaded revised version. The new content is marked in blue.
>
> *Q2*: Are there any assumptions on the type of kernel F$^3$M is useful for? It seems like most experiments were performed on RBF kernels.
> *A2*: Not really, but the method generally extends directly to translation invariant kernels.
>
> *Q3*: Line 143 should $u_1$  and $u_2$  be $v_1$  and $v_2$?
> *A3*: Good spot! Thanks, we have corrected it.
>
> *Q4*: Can the authors include more detail in Section 4.1 on the low-level GPU optimizations performed in the main text?
> *A4*:  Yes! We will extend the paragraphs about low-level implementations by including more pseudo-code and more details about the algorithmic challenges encountered during this project. As these changes require more careful writing, we have decided to not include them in a rebuttal revision in hopes of carefully rewriting this section in a more comprehensive way.

---

> > ### Comment · Reviewer_TBZE · 2022-08-08
> > **Response**
> >
> > I'm grateful to the authors for their response to my questions. I will keep my score as accept.

---

### Author Response · Authors · 2022-07-30
**General response**

We would like to thank the reviewers for their comments, we believe they have significantly improved the work. We are happy that the reviewers find the problem interesting and the method to be of significance. We respond to each reviewer's comments individually.

---

### Meta-Review · Area_Chair_EgnN · 2022-08-23

**Recommendation:** Accept
**Confidence:** Less certain

**Metareview:**

The authors propose an new approximation procedure for Kernel matrix-vector multiplication target to tall and skinny kernel matrices. The proposed method achieves significant speedups over the state-of-the-art GPU-based linear solver FALKON while sacrificing only small drops in accuracy due to approximation.

The paper discusses a specific use case (low dimensional data) but it is very clear about the scope. The reviewers agree that the problem still has high significance, is well motivated and the reported performance gains are convincing. The experiments also provide interesting insights into the inner working of the method and the trade-offs between accuracy and efficiency.

For a potential camera ready version the authors should carefully work in the reviewers' comments on the presentation to improve the accessibility of their work for a general NeurIPS audience. Also additional details about the low-level GPU optimizations would be good to add in section 4.1 and some comments on how to extend the method to other kernel functions would strengthen the paper.

**Award:**

No

---

### Decision · Program_Chairs · 2022-09-14

Accept